# Statins, metformin, and RAS inhibitors did not reduce variceal bleeding risk and mortality in a large, real-life cohort of patients with cirrhosis

**Nikolaus Pfisterer**[1,2,3], **Michael Schwarz**[1,2], **Caroline Schwarz**[1,2,4], **Florian Putre**[1,2], **Lukas Ritt**[1,2], **Florian Riedl**[1,2], **Lukas Hartl**[1,2], **Mathias Jachs**[1,2], **Mattias Mandorfer**[1,2], **Christian Madl**[3,5], **Michael Trauner**[1], **Thomas Reiberger**[1,2,6]*

1 Division of Gastroenterology and Hepatology, Department of Medicine III, Medical University of Vienna, Vienna, Austria, 2 Vienna Hepatic Hemodynamic Lab, Medical University of Vienna, Vienna, Austria, 3 Klinik Landstrasse, 4. Medizinische Abteilung für Gastroenterologie und Hepatologie, Vienna, Austria, 4 Klinik Ottakring, 4. Medizinische Abteilung für Gastroenterologie und Hepatologie, Wien, Austria, 5 Sigmund Freud University, Private Medical School, Vienna, Austria, 6 Christian-Doppler Laboratory for Portal Hypertension and Liver Fibrosis, Medical University of Vienna, Vienna, Austria

* thomas.reiberger@meduniwien.ac.at

**Data Availability Statement:** All relevant data are within the paper.

## Abstract

### Background

Previous experimental and clinical studies suggested a beneficial effect of statins, metformin, angiotensin-converting-enzyme inhibitors and angiotensin II receptor blockers (RASi) on portal hypertension. Still, their effects on hard cirrhosis-related clinical endpoints, such as variceal bleeding and bleeding-related mortality, remain to be investigated.

### Methods

Thus, we recorded the use of statins, metformin and RASi in a large cohort of cirrhotic patients undergoing endoscopic band ligation (EBL) for primary (PP, n = 440) and secondary bleeding prophylaxis (SP, n = 480) between 01/2000 and 05/2020. Variceal (re-) bleeding and survival rates were compared between patients with vs. without these co-medications.

### Results

A total of 920 cirrhotic patients with varices were included. At first EBL, median MELD was 13 and 515 (56%) patients showed ascites. Statins, metformin and RASi were used by 49 (5.3%), 74 (8%), and 91 (9.9%) patients, respectively. MELD and platelet counts were similar in patients with and without the co-medications of interest. Rates of first variceal bleeding and variceal rebleeding at 2 years were 5.2% and 11.7%, respectively. Neither of the co-medications were associated with decreased first bleeding rates (log-rank tests in PP: statins p = 0.813, metformin p = 0.862, RASi p = 0.919) nor rebleeding rates (log-rank tests in SP: statin p = 0.113, metformin p = 0.348, RASi p = 0.273). Similar mortality rates were

**Funding:** The author(s) received no specific funding for this work.

**Competing interests:** The authors have declared that no competing interests exist.

documented in patients with and without co-medications for PP (log-rank tests: statins p = 0.630, metformin p = 0.591, RASi p = 0.064) and for SP (statins p = 0.720, metformin p = 0.584, RASi p = 0.118).

## Conclusion

In clinical practice, variceal bleeding and mortality rates of cirrhotic patients were not reduced by co-medication with statins, metformin or RASi. Nevertheless, we recommend the use of these co-medications by indication, as they may still exert beneficial effects on non-bleeding complications in patients with liver cirrhosis.

## Introduction

Despite improvements in the management of patients with liver cirrhosis, variceal bleeding remains a severe complication of portal hypertension and is still associated with a high mortality rate of up to 20% [1–3]. Importantly, in survivors of acute variceal bleeding, the rebleeding rate is about 60% and the mortality rate rises to 33% [4]. Therefore, primary and secondary prophylactic treatment of portal hypertension is of paramount importance to avoid variceal (re-)bleeding. Current guidelines recommend either non-selective betablockers (NSBBs) or endoscopic band ligation (EBL) for primary prophylaxis, while combined treatment with NSBB and EBL is recommended as a secondary prophylaxis for variceal (re-)rebleeding [5–14]. Despite these established treatments, there is still a need for more effective treatments to manage patients with portal hypertension. Of note, approximately 15% of patients have NSBB intolerance while more than 50% of the patients achieve optimal HVPG response of NSBB [14–16]. Therefore, repurposing of approved drugs for which experimental and translational studies have suggested some benefit regarding improvement of portal hypertension, hepatic function, or liver-related outcomes has been suggested.

Both preclinical and clinical evidence suggest a potential benefit of statins, metformin, angiotensin-converting-enzyme inhibitors, and angiotensin II receptor blockers (RASi) on portal hypertension, as will be further elaborated as follows [17–23].

Statins–commonly used as lipid-lowering drugs - inhibit the activity of 3-hydroxy-3-methylglutarylcoenzym A reductase and thereby decrease oxidative stress and inflammation by enhancing endothelial nitric oxide synthase (eNOS) activity. Statin-induced eNOS activity can promote intrahepatic vasodilatation and thus, lower portal pressure [20, 21, 24]. Additionally, the antifibrotic effects of statins have been demonstrated by reduced collagen- and cytokine production in hepatic stellate cells [25]. Recent studies suggested a beneficial impact of stations on portal hypertension as well as on survival [21, 26, 27]. Still, there is no sufficient evidence to generally recommend statins for the treatment of portal hypertension in patients with cirrhosis [28, 29].

Insulin sensitizers were described to lead to NO enhancement, reduce oxidative stress and alterations in vascular smooth muscle cells, and inhibit the α-adrenergic tone in patients with diabetes mellitus [22, 30–33]. In type-2 diabetes patients, metformin was shown to have an impact on the mesenteric artery by suppressing prostanoids and improving vascular resistance in animal studies [31].

Therefore, a decrease in portal pressure in patients with liver cirrhosis using metformin seems reasonable, but the efficacy and safety of metformin for treating complications associated with portal hypertension have not yet been assessed in formal clinical trials. However,

one retrospective study showed that using metformin was significantly associated with a 57% reduction in the risk of all-cause mortality in patients with liver cirrhosis and diabetes. In one animal study, metformin had an additive effect with propranolol for a significant reduction of portal hypertension [22]. Furthermore, cirrhotic patients with diabetes were shown to have a higher incidence of ascites, renal dysfunction, bacterial infections, and hepatic encephalopathy [34].

Angiotensin II and aldosterone promote intrahepatic resistance, inflammation, endothelial dysfunction, and deposition of fibrous tissue [17, 19, 23, 35]. Conversely, (pharmacological) RAS inhibition may lead to a decrease in portal pressure [17, 18, 35]. Interestingly, one meta-analysis involving n = 678 patients demonstrated that the use of angiotensin-converting enzyme inhibitors (ACEi) or angiotensin II receptor blockers (ARB) resulted in a reduction in portal pressure among patients with Child-Pugh A cirrhosis, without reported adverse events [23]. However, studies assessing the impact of RASi on portal hypertension still show conflicting results. Furthermore, no clinical data on variceal bleeding is currently available [36, 37].

Esophageal variceal bleeding is a severe complication of portal hypertension. However, current data on the drugs mentioned above and their impact on the risk of variceal bleeding is scarce. Therefore, we designed a retrospective clinical multicenter study to assess the influence of statins, metformin, and RASi on the rates of variceal bleeding and mortality.

## Patients and methods

### Study design

In this retrospective study conducted at two tertiary clinical centers (Vienna General Hospital of the Medical University of Vienna and Klinik Landstrasse in Vienna), we assessed patients with liver cirrhosis and portal hypertension in whom esophageal varices were detected during endoscopy. Patients over the age of 18 years undergoing elective endoscopic band ligation (EBL) for esophageal varices between the 1st of January 2000 and the 31st of May 2020 were included.

The data were initially accessed in February 2016 using the keyword "Varize/n" ("varice/s" in German) through the endoscopy software (ViewPoint, Version 5.6.27.232, GE Health) and the hospital electronic records database (impuls.kis, version 4.0.3, systema). Furthermore, the data were updated and accessed again in July 2020. All parameters in the follow-up were extracted from the hospital electronic records database (impuls.kis, version 4.0.3, systema).

Patients with non-cirrhotic portal hypertension receiving EBL/endoscopic treatment, those with previous transjugular intrahepatic portosystemic shunt (TIPS) implantation or orthotopic liver transplantation, patients with hepatocellular carcinoma who did not fulfill the Milan criteria, those who had occlusive portal vein thrombosis, patients < 18 years of age, and patients with insufficient medical/endoscopic records were excluded from the study. (Re-)bleeding rates and transplant-free survival were investigated as primary endpoints. We compared patients with liver cirrhosis who underwent EBL for esophageal varices, and assessed the effects of concomitant treatment with statins, metformin, and/or RASi. It is important to note that the decision to use these drugs was based on the treating physician's discretion according to indication, comorbidities, and tolerability considering comedication.

### Parameters

Clinical (age, sex, etiology of cirrhosis, presence and grade of ascites, and presence and grade of hepatic encephalopathy) and laboratory parameters (aspartate transaminase, alanine transaminase, gamma-glutamyl transferase, serum-bilirubin, prothrombin time, international normalized ratio, and platelet count), as well as specific data on clinical outcomes for our primary outcomes,

such as death, variceal (re-)bleeding, TIPS implantation, and orthotopic liver transplantation were recorded. Follow-up time was defined until death or until the last clinical visit for all patients. Variceal (re-)bleeding was defined as the presence of hematemesis and/or clinical and laboratory evidence of acute blood loss from esophageal varices. Laboratory parameters, size of varices, number of gastroscopies and endoscopic band ligations (EBL) per patient, presence of ascites, concomitant non-selective betablockers (NSBB) prescriptions and number of bleeding episodes were recorded. The MELD (model for end-stage liver disease) score and the Child-Pugh score were calculated to estimate the severity of liver disease for the study population [38, 39].

## Statistics

For statistical analyses, IBM SPSS statistics Version 28 (SPSS Inc., Armonk, New York, USA) and GRAPHPAD Prism 9 (GRAPHPAD Software, La Jolla, California, USA) were used. Continuous variables were reported as median (range) and categorical variables were shown as numbers (n). Proportions (%) of patients with certain characteristics were shown. Patient characteristics were evaluated according to statin, metformin, and RASi use as well as one cohort without co-medications. Comparisons of continuous variables were performed using Student t-test or Mann-Whitney U-test, if applicable. Comparisons of categorical variables were performed using Chi-square or Fisher's exact test. A Cox regression model was used to assess the effect of variables such as age, Child-Pugh stage, and the use of co-medications on (re-)bleeding and death. Clinical outcomes (death and variceal bleeding) were analyzed by Kaplan-Meier curves, and log-rank tests for group comparisons were performed. A p-value ≤0.05 was considered statistically significant.

## Ethics and data availability statement

This retrospective multicenter study was managed in accordance with the Declaration of Helsinki and approved by the ethics committee of the Medical University of Vienna (EK#1666/2015) and the ethics committee of the Wiener Gesundheitsverbund in Vienna (MA-15, EK#15-280-VK).

## Results

### Patient characteristics (Fig 1 and Tables 1–3)

A total of 1257 patients underwent endoscopic treatment for gastroesophageal varices in our two tertiary care centers. A total number of n = 204 patients were excluded due to noncirrhotic portal hypertension, death at first endoscopy, insufficient endoscopic and/or medical records, hepatocellular carcinoma outside the Milan criteria, implantation of a transjugular intrahepatic portosystemic shunt (TIPS) or orthotopic liver transplantation (OLT) prior to the study, and occlusive portal vein thrombosis. Finally, among 920 included patients with varices, statins, metformin and RASi were used in 49 (5.3%), 74 (8%), and 91 (9.9%), respectively.

Overall, the median Model for End-Stage Liver Disease (MELD) score at the first endoscopic band ligation (EBL) was 13 (interquartile range [IQR]: 16), with n = 515 (56%) of the patients exhibiting ascites. Among the n = 440 (47.8%) patients undergoing EBL for primary prophylaxis, n = 32 (7.3%) received statins, n = 40 (9.1%) received metformin, n = 61 (13.9%) received angiotensin-converting-enzyme inhibitors or angiotensin II receptor blockers (RASi). During a median follow-up of 22.6 months, n = 58/440 (13.2%) patients suffered first variceal bleeding and n = 162 (36.8%) patients died.

Among the n = 480 (52.2%) patients who were treated with EBL for secondary prophylaxis, n = 17 (3.5%) received statins, n = 34 (7.1%) received metformin, and n = 30 (6.3%) received

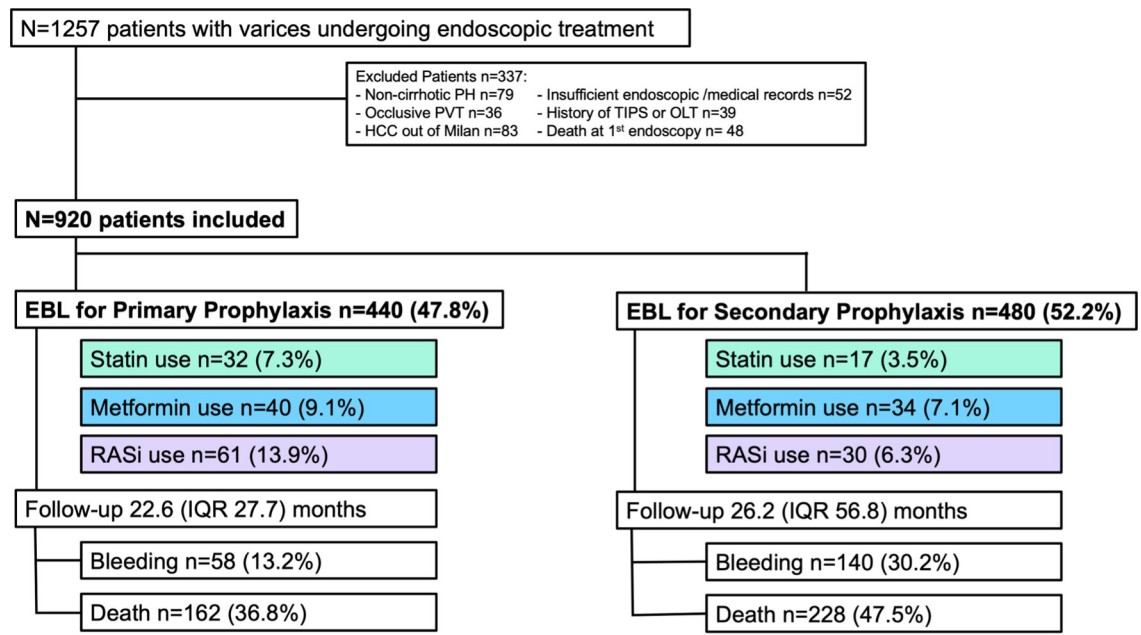

**Fig 1. Patient flow chart.** Among 1257 patients undergoing endoscopic treatment, 920 patients with esophageal varices were finally included in this study. These patients were divided into groups of primary or secondary prophylaxis and according to the concomitant treatment with statin, metformin and RASi. Abbreviations: N (total numbers), IQR (interquartile range), PHT (portal hypertension), OLT (orthotopic liver transplantation), TIPS (transjugular intrahepatic portosystemic shunt). PVT (portal vein thrombosis) Continuous variables are expressed as median with interquartile range (IQR).

RASi. Over a median follow-up time of 26.2 months, n = 140/480 (30.2%) patients in secondary prophylaxis had further variceal bleeding events and n = 228 (47.5%) died.

The differences in outcomes between primary prophylaxis and secondary prophylaxis, including bleeding during follow-up and death during follow-up, are statistically significant with p<0.001 and p = 0.001, respectively.

Furthermore, in the primary prophylaxis group, n = 99 (22.7%) of patients were on at least one comedication (statin, metformin, or RASi), while in the secondary prophylaxis group was significantly lower with n = 68 (14.7%, p = 0.026). For this analyzing, patients in primary prophylaxis (n = 4) and secondary prophylaxis (n = 18) were not considered due to insufficient information on these co-medications.

The median follow-up duration was similar between groups, with primary prophylaxis at 26.8 months (interquartile range: IQR 37.9) and secondary prophylaxis at 29.2 months (IQR 56.7, p = 0.782). However, patients in secondary prophylaxis showed a significantly higher incidence of bleeding during follow-up (29.2%) compared to patients in primary prophylaxis (13.2%, p<0.001). Mortality rates also were higher in the secondary group (47.5%) than in the primary group (36.8%, p = 0.001). Additionally, one-year outcomes demonstrated significant differences in both bleeding (p = 0.012) and mortality rates (p = 0.026). In the analysis of co-medication usage, a significantly higher percentage of patients in the primary prophylaxis group, 22.7%, were found to be on at least one co-medication, in contrast to only 14.7% in the secondary prophylaxis group (p = 0.026).

The characteristics of patients undergoing EBL for primary or secondary prophylaxis were assessed according to statin use, metformin use, RASi use, and absence of these mentioned co-medications. Patients in whom co-medication could not be reliably assessed were not considered in this analysis.

**Table 1. Characteristics of patients undergoing endoscopic band ligation for primary bleeding prophylaxis according to intake of co-medication of interest.**

| Primary prophylaxis (n = 440) | No comeds | Statin | Metformin | RASi |
|---|---|---|---|---|
| | 337 (76.6%) | 32 (7.3%) | 40 (9.1%) | 61 (13.9%) |
| Age (median, IQR) | 57.2 (15.8) | 66.3 (15.8) | 65.2 (17.6) | 66.3 (13.2) |
| Sex (m/w, %m) | 225/112 (66.8) | 19/13 (59.4) | 27/13 (67.5) | 33/28 (54.1) |
| Aetiology of cirrhosis | | | | |
| • Alcohol (n, %) | 169 (50.1) | 14 (43.8) | 17 (42.5) | 27 (44.3) |
| • Viral hepatitis (n, %) | 71 (21.1) | 5 (15.6) | 8 (20) | 15 (24.6) |
| • Mixed (n, %) | 15 (4.4) | 0 (0) | 0 (0) | 0 (0) |
| • Other (n, %) | 41 (12.2) | 3 (9.3) | 2 (5) | 2 (3.3) |
| • Cryptogenic (n, %) | 41 (12.2) | 10 (31.3) | 13 (32.5) | 17 (27.9) |
| Gastric varices (n, %) | 5 (1.5) | 0 (0) | 1 (2.5) | 1 (1.6) |
| Small varices <5mm (n, %) | 35 (10.4) | 1 (3.1) | 3 (7.5) | 6 (9.8) |
| Large varices >5mm (n, %) | 290 (86.1) | 30 (93.8) | 36 (90) | 52 (85.2) |
| Serum creatinine (mg/dL, IQR) | 0.83 (0.31) | 0.79 (0.40) | 0.74 (0.25) | 0.93 (8.60) |
| Serum albumine (mg/dL, IQR) | 33.4 (8.1) | 35.9 (12.5) | 36.1 (9.5) | 36 (8.6) |
| PT (%, IQR) | 59 (23) | 68 (24.5) | 68 (20.5) | 66 (17) |
| INR (IQR) | 1.3 (0.3) | 1.2 (0.3) | 1.2 (0.2) | 1.2 (0.3) |
| Serum-bilirubin (mg/dL, IQR) | 1.7 (2) | 1 (1.8) | 1.1 (1.4) | 1.1 (1.1) |
| MELD (IQR) | 12 (6) | 10 (5) | 9 (5.5) | 10 (5) |
| Ascites | | | | |
| • mild-moderate (n, %) | 137 (40.7) | 11 (34.4) | 13 (3.3) | 21 (34.4) |
| • severe/refractory (n, %) | 71 (21.1) | 4 (12.5) | 6 (15) | 6 (9.8) |
| Child-Pugh stage | | | | |
| • CPS A (n, %) | 93 (27.6) | 12 (37.8) | 18 (45) | 31 (50.8) |
| • CPS B (n, %) | 111 (32.9) | 7 (21.9) | 12 (30) | 16 (26.2) |
| • CPS C (n, %) | 60 (17.8) | 4 (12.5) | 1 (2.5) | 4 (6.6) |
| NSBB use (n, %) | 231 (68.6) | 22 (68.8) | 33 (82.5) | 46 (75.4) |
| AST (U/L, IQR) | 46.5 (51) | 36 (33.3) | 36 (14.5) | 41 (22) |
| ALT (U/L, IQR) | 30 (27) | 26 (17.3) | 28 (19.5) | 25.5 (19.5) |
| GGT (U/L, IQR) | 90.5 (126.6) | 121 (187.1) | 90 (118) | 108 (93) |
| Platelets (G/L, IQR) | 89 (75.5) | 106 (105) | 115.5 (108.3) | 109 (74.6) |
| Median follow-up (months, IQR) | 27.1 (41.4) | 25.4 (26.9) | 24.9 (25.5) | 24.2 (29.9) |
| Number of endoscopies (per year/IQR) | 2.8 (6.9) | 3 (5.3) | 4 (5) | 3.5 (5.8) |
| Number of EBL sessions (per year/IQR) | 1.3 (4.3) | 1.7 (2.3) | 12.1 (2.7) | 1.7 (4.8) |
| First variceal bleeding (n, %) | 45 (13.4) | 2 (6) | 5 (12.5) | 6 (9.8) |
| OLT (n, %) | 33 (9.8) | 0 (0) | 4 (10) | 2 (3.3) |
| Death (n, %) | 116 (34.4) | 12 (37.5) | 16 (40) | 27 (44.3) |

Abbreviations: RASi (Angiotensin-converting-enzyme inhibitors and angiotensin II receptor blockers), Comeds (Comedication included: statin, metformin and RASi), NSBB (non-selective betablockers), IQR (interquartile range), m (male), f (female), mm (millimeters), n (total numbers), mg/dL (milligram per deciliter), PT (prothrombin time), INR (International Normalized Ratio), MELD (Model for End-Stage Liver Disease), AST (Aspartate transaminase), ALT (Alanine transaminase), GGT(Gamma-glutamyl transferase), U/L (unit per liter), CPS (Child-Pugh stadium)

In primary prophylaxis, patients with co-medications were older as compared to those without co-medications (statin: 66.3 IQR 15.8 years vs. metformin: 65.2 IQR 17.6 years vs. RASi: 66.3 IQR 13.2 years vs. non-co-medications: 57.2 IQR 66.8). Higher serum creatinine values were seen in patients with RASi as compared to those with other or no co-medications

**Table 2. Characteristics of patients undergoing endoscopic band ligation for secondary bleeding prophylaxis according to co-medications.**

| Secondary prophylaxis (n = 480) | No comeds | Statin | Metformin | RASi |
|---|---|---|---|---|
| | 394 (82.1%) | 17 (3.5%) | 34 (7.1%) | 30 (6.3%) |
| Age (median, IQR) | 53.9 (14.9) | 59.7 (17.6) | 62.3 (16.9) | 61.8 (23.4) |
| Sex (m/w, %m) | 276/118 (70.1) | 8/9 (47.1) | 26/8 (76.5) | 19/11 (6.3) |
| Aetiology of cirrhosis | | | | |
| • Alcohol (n, %) | 224 (56.9) | 9 (52.9) | 14 (41.2) | 10 (33.3) |
| • Viral hepatitis (n, %) | 81 (20.6) | 1 (5.9) | 6 (17.6) | 4 (13.3) |
| • Mixed (n, %) | 27 (6.9) | 0 (0) | 3 (8.8) | 0 (0) |
| • Other (n, %) | 24 (6.1) | 4 (23.6) | 4 (11.8) | 7 (23.4) |
| • Cryptogenic (n, %) | 38 (9.6) | 3 (17.6) | 7 (20.6) | 9 (30) |
| Gastric varices (n, %) | 10 (2.5) | 1 (5.9) | 2 (5.9) | 0 (0) |
| Small varices <5mm (n, %) | 53 (13.5) | 1 (5.9) | 4 (11.8) | 4 (13.3) |
| Large varices >5mm (n, %) | 302 (76.6) | 15 (88.2) | 29 (85.3) | 24 (80) |
| Serum-creatinine (mg/dL, IQR) | 0.90 (0.40) | 0.93 (0.34) | 0.93 (0.31) | 1.10 (0.40) |
| Serum-albumine (mg/dL, IQR) | 31.5 (9) | 32.8 (12.2) | 32.5 (12) | 37.2 (10.2) |
| PT (%, IQR) | 55 (21) | 63 (46.5) | 68 (25) | 63 (31) |
| INR (IQR) | 1.4 (0.4) | 1.2 (0.7) | 1.2 (0.3) | 1.3 (0.2) |
| Serum-bilirubin (mg/dL, IQR) | 1.9 (2.1) | 1.4 (1.2) | 1.5 (1.7) | 1.2 (1.1) |
| MELD (IQR) | 13 (7) | 14 (6) | 11 (5.8) | 12 (5.5) |
| Ascites | | | | |
| • none (n, %) | 170 (43.1) | 9 (52.9) | 16 (47.1) | 18 (60) |
| • mild-moderate (n, %) | 144 (36.5) | 8 (47.1) | 11 (32.3) | 10 (33.3) |
| • severe/refractory (n, %) | 76 (19.3) | 0 (0) | 5 (14.7) | 2 (6.6) |
| Child-Pugh stage | | | | |
| • CPS A (n, %) | 92 (23.4) | 5 (29.4) | 9 (26.5) | 15 (50) |
| • CPS B (n, %) | 180 (45.7) | 6 (35.3) | 13 (38.2) | 6 (20) |
| • CPS C (n, %) | 89 (22.6) | 1 (5.9) | 4 (11.8) | 4 (13.3) |
| NSBB use (n, %) | 272 (69) | 14 (82.4) | 25 (73.5) | 18 (60) |
| AST (U/L, IQR) | 52 (59) | 57.5 (30.8) | 35.5 (22) | 51 (36.5) |
| ALT (U/L, IQR) | 32 (28.8) | 36 (19) | 31.5 (30.5) | 31 (21) |
| GGT (U/L, IQR) | 110 (188.8) | 132 (395.5) | 85 (147.5) | 88 (147.8) |
| Platelets (G/L, IQR) | 101 (89) | 131 (41) | 91 (62) | 63 (31) |
| Median FU (months, IQR) | 30.3 (60.9) | 27.3 (23.5) | 25.2 (42.5) | 26.4 (46.8) |
| Number of endoscopies (per year/IQR) | 1.8 (7.8) | 5.9 (22.4) | 4.4 (10.4) | 2.3 (7) |
| Number of EBL sessions (per year/IQR) | 1 (6.3) | 4.1 (18.3) | 2.2 (6.8) | 2.2 (4.9) |
| Variceal Re-bleeding (n, %) | 108 (27.4) | 8 (47.1) | 10 (29.4) | 9 (30) |
| OLT (n, %) | 41 (10.4) | 1 (5.9) | 1 (2.9) | 2 (6.7) |
| Death (n, %) | 174 (44.2) | 7 (41.2) | 17 (50) | 19 (63.3) |

Abbreviations: RASi (Angiotensin-converting-enzyme inhibitors and angiotensin II receptor blockers), Comeds (Comedication included: statin, metformin and RASi), NSBB (non-selective betablockers), IQR (interquartile range), m (male), f (female), mm (millimeters), n (total numbers), mg/dL (milligram per deciliter), PT (prothrombin time), INR (International Normalized Ratio), MELD (Model for End-Stage Liver Disease), AST (Aspartate transaminase), ALT (Alanine transaminase), GGT(Gamma-glutamyl transferase), U/L (unit per liter), CPS (Child-Pugh stadium)

(statin: 0.79mg/dL IQR 0.40 vs. metformin: 0.74mg/dL IQR 0.25 vs. RASi: 0.93mg/dL IQR 8.60 vs. non-co-medications: 0.83mg/dL IQR 0.31).

Lower albumin levels were seen in patients without co-medications (statin: 35.9mg/dL IQR 12.5 vs. metformin: 36.1mg/dL IQR 9.5 vs. RASi: 36mg/dL IQR 8.6 vs. non-co-medications:

**Table 3. Comparison of co-medication usage in primary vs. secondary prophylaxis and Child-Pugh A vs. Child-Pugh B/C.**

|  | Primary prophylaxis | Secondary prophylaxis | p-value |
|---|---|---|---|
| Follow-up (months, IQR) | 26.8 (37.9) | 29.2 (56.7) | p = 0.782 |
| Bleeding during Follow up (n,%) | 58 (13.2) | 140 (29.2) | p<0.001 |
| Bleeding at one year (n,%) | 17 (3.9) | 38 (7.9) | p = 0.012 |
| Death during Follow up (n,%) | 162 (36.8) | 228 (47.5) | p = 0.001 |
| Death at one year (n,%) | 60 (13.6) | 92 (19.2) | p = 0.026 |
| At least one Comed (n,%) | 99 (22.7) | 68 (14.7) | p = 0.026 |
| No Comeds (n,%) | 337 (77.3) | 394 (85.3) | |
|  | **Child A** | **Child B/C** | |
| At least one Comed (n,%) | 70 (27.5) | 62 (12.3) | p<0.001 |
| No comeds (n,%) | 185 (72.5) | 440 (87.6) | |

Abbreviations: Comeds (Comedication included: statin, metformin and RASi), RASi (Angiotensin-converting-enzyme inhibitors and angiotensin II receptor blockers), IQR (interquartile range), n (total numbers), CPS (Child-Pugh stadium)

33.4 IQR 8.1). Importantly, median MELD score (statin: 10 IQR 5 vs. metformin: 9 IQR 5.5 vs. RASi: 10 IQR 5 vs. non-co-medications: 12 IQR 6) and the rate of Child C (statin: 12.5% vs. metformin: 2.5% vs. RASi: 6.6% vs. non-co-medications: 17.8%) were higher in patients without co-medications as compared to the other groups. Furthermore, patients without co-medications had higher AST levels (statin: 36U/L IQR 33.3 vs. metformin: 36U/L IQR 14.5 vs. RASi: 41 IQR 22 non-co-medications: 46.5 IQR 51). The numbers of performed endoscopies per year (statin: 3 IQR 5.3 vs. metformin: 4 IQR 5 vs. RASi: 3.5 IQR 5.8 vs. non-co-medications: 2.8 IQR 6.9) and EBLs per year (statin: 1.7 IQR 2.3 vs. metformin: 12.1 IQR 2.7 vs. RASi: 1.7 IQR 4.8 vs. non-co-medications: 1.3 IQR 4.3) were higher among individuals with co-medications. Orthotopic liver transplantation (OLT) was not performed in any patients using metformin (statin: 33 9.8% vs. metformin: 0 vs. RASi: 4 10% vs. non-co-medications: 2 3.3%).

In secondary prophylaxis, patients with co-medications were younger than those without co-medications (statin: 59.7.3 IQR 17.6 years vs. metformin: 62.3 IQR 16.9 years vs. RASi: 61.8 IQR 23.4 years vs. non-co-medications: 53.9 IQR 14.9). Similar to primary prophylaxis, higher serum creatinine values were also observed for patients with RASi in secondary prophylaxis (statin: 0.93mg/dL IQR 0.34 vs. metformin: 0.93mg/dL IQR 0.31 vs. RASi: 1.10mg/dL IQR 0.40 vs. non-co-medications: 0.90mg/dL IQR 0.40). However, in secondary prophylaxis, higher albumin levels were seen in patients receiving RASi (statin: 32.8mg/dL IQR 12.2 vs. metformin: 32.5mg/dL IQR 12 vs. RASi: 37.2mg/dL IQR 10.2 vs. non-co-medications: 31.5 IQR 9).

Patients within Child-Pugh stages B/C received significantly fewer co-medications (at least one of the co-medications: Child-Pugh A: 27.5% vs. Child-Pugh B/C: 12.3%, p<0.001). Child C cirrhosis was still more frequent among individuals without co-medications as compared to the other groups in secondary prophylaxis (statin: 5.9% vs. metformin: 11.8% vs. RASi: 13.3% vs. non-co-medications: 22.6%). AST levels were lower in patients without co-medications compared to those with co-medications (statin: 57.5U/L IQR 30.8 vs. metformin: 35.5U/L IQR 22 vs. RASi: 51U/L IQR 36.5 vs. non-co-medications: 52 U/L IQR 59). In primary prophylaxis as well as secondary prophylaxis, the numbers of performed endoscopies per year (statin: 5.9 IQR 22.4 vs. metformin: 4.4 IQR 10.4 vs. RASi: 2.3 IQR 7 vs. non-co-medications: 1.8 IQR 7.8) and EBLs per year (statin: 4.1 IQR 18.3 vs. metformin: 2.2 IQR 6.8 vs. RASi: 2.2 IQR 4.9 vs. non-co-medications: 1 IQR 6.3) were higher in patients with co-medications. In the setting of secondary prophylaxis, more liver transplantations (10.4%) were performed in the group of patients without co-medications.

## Bleeding rates for all patients with co-medication (statin, metformin or RASi) or non-co-medication (Fig 2 and Table 4)

For all patients in this study, concomitant therapy with a statin (Hazard ratio HR: 1.47; confidence interval 95%CI: 0.71–3.03; log-rank p = 0.213; *Fig 2A*), metformin (HR:1.20; 95%CI: 0.68–2.12; log-rank p = 0.496; *Fig 2B*), or RASi (HR: 1.09; 95%CI 0.63–1.87; log-rank p = 0.758; *Fig 2C*) did not show an impact on (re-)bleeding rates as compared to absence of these co-medications. Bleeding rates in primary and secondary prophylaxis are shown in Fig 2D and 2E and were not significantly influenced by co-medication status in Year 2. Generally, the intake of at least one co-medication did not show a significant impact on (re-)bleeding risk, even after adjustment for Child-Pugh score and age (odds ratio, OR: 1.02, confidence interval 95%CI 0.63–1.65, p = 0.945; see Table 4).

## Mortality for all patients with co-medication (statin, metformin or RASi) or non-co-medication (Fig 3 and Table 4)

The transplant-free survival rate for all patients was not significantly decreased by the addition of statin (HR:1.01; 95%CI: 0.62–1.62; log-rank p = 0.978), metformin (HR: 1.14; 95%CI: 0.77–1.68; log-rank p = 0.488) or RASi (HR: 1.42; 95%CI: 0.99–2.04; log-rank p = 0.767). Mortality rates in primary and secondary prophylaxis are shown in Fig 3D and 3E and were not significantly influenced by co-medication status in Year 2. Furthermore, for all patients in the multivariate analysis, the intake of at least one co-medication, even after adjustment for age and Child B/C status, did not significantly impact survival outcomes (OR: 1.39, 95%CI 0.92–2.09, p = 0.115, see Table 4).

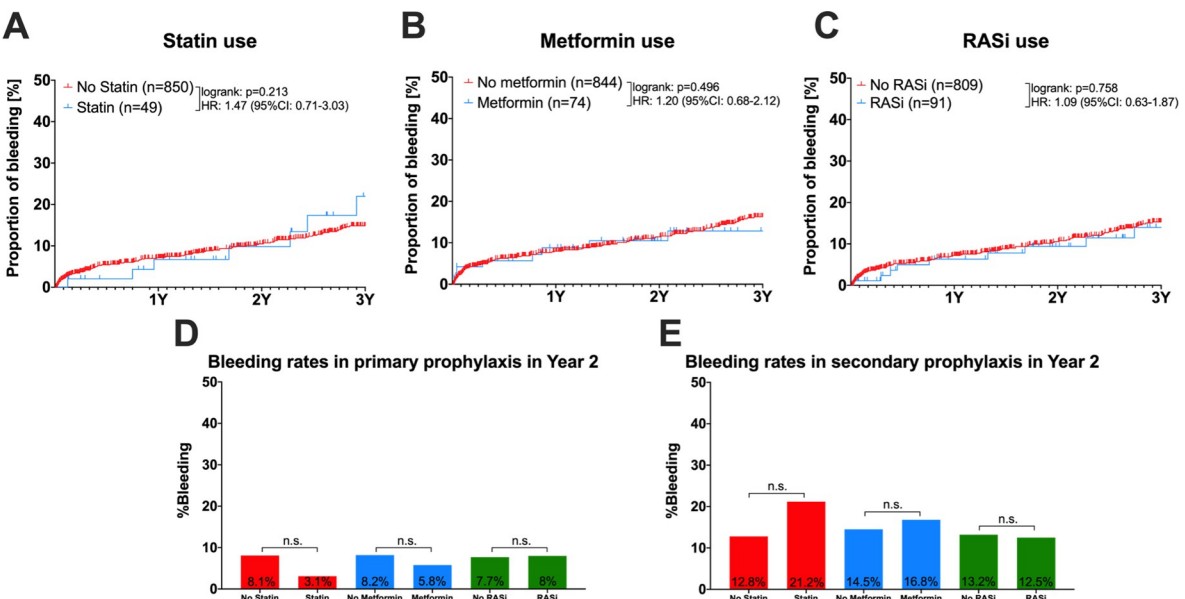

**Fig 2. Rates of variceal bleeding.** Kaplan-Meier curves of (re-)bleeding rates for all patients with comedication (statin, metformin or RASi) or without comedication (A-C). Bar chart of first bleeding rates (D) and variceal rebleeding rates (E) within 2 years after first endoscopic band ligation, grouped by intake vs. non-intake of the respective co-medication of interest. n (total numbers), 1Y (Year 1), 2Y (Year 2), 3Y (Year 3), HR (Hazard-Ratio); 95% CI (95% confidence interval); n.s. (not significant); statistical comparisons were performed by log-rank tests.

**Table 4. Risk of (re-)bleeding and death using at least one comedication.**

| | No bleeding | (Re-)bleeding | Multivariate odds ratio (95% CI) | p-value |
|---|---|---|---|---|
| **Patients (n)** | **710** | **187** | | |
| **Age (median, IQR)** | 57.6 (16.2) | 55.5 (17.1) | 0.99 (0.98–1.00) | p = 0.157 |
| **Child-Stage A vs. B/C (n, %B/C)** | 204/400 (55.4) | 54/90 (45.5) | 1.00 (0.69–1.46) | p = 0.999 |
| **No comed vs. Comed (n,%Comed)** | 578/132 (18.6) | 153/34 (18.2) | 1.02 (0.63–1.65) | p = 0.945 |
| | No death | Death | Multivariate odds ratio (95% CI) | p-value |
| **Patients (n)** | **529** | **369** | | |
| **Age (median, IQR)** | 55.8 (15.9) | 59.7 (16.7) | 1.03 (1.01–1.04) | p<0.001 |
| **Child-Stage A vs. B/C (n, %B/C)** | 178/268 (50.7) | 80/249 (67.5) | 2.33 (1.66–3.26) | p<0.001 |
| **No comed vs. Comed (n,%Comed)** | 441/88 (16.6) | 290/79 (21.4) | 1.39 (0.92–2.09) | p = 0.115 |

Abbreviations: Comeds (Comedication included: statin, metformin and RASi), RASi (Angiotensin-converting-enzyme inhibitors and angiotensin II receptor blockers), IQR (interquartile range), n (total numbers)

## Mortality for patients with co-medication (statin, metformin or RASi) or non-co-medication in primary and secondary prophylaxis (Fig 4)

In *Fig 4*, patients were subdivided according to primary and secondary prophylaxis and the impact of co-medication status on mortality was analyzed. In this real-life cohort, statin intake rate did not significantly impact mortality in primary prophylaxis (HR: 1.16; 95%CI: 0.62–2.17; log-rank p = 0.630) and secondary prophylaxis (HR: 0.86; 95%CI: 0.40–1.84; log-rank p = 0.720). Transplant-free survival did not decrease significantly when metformin was used in

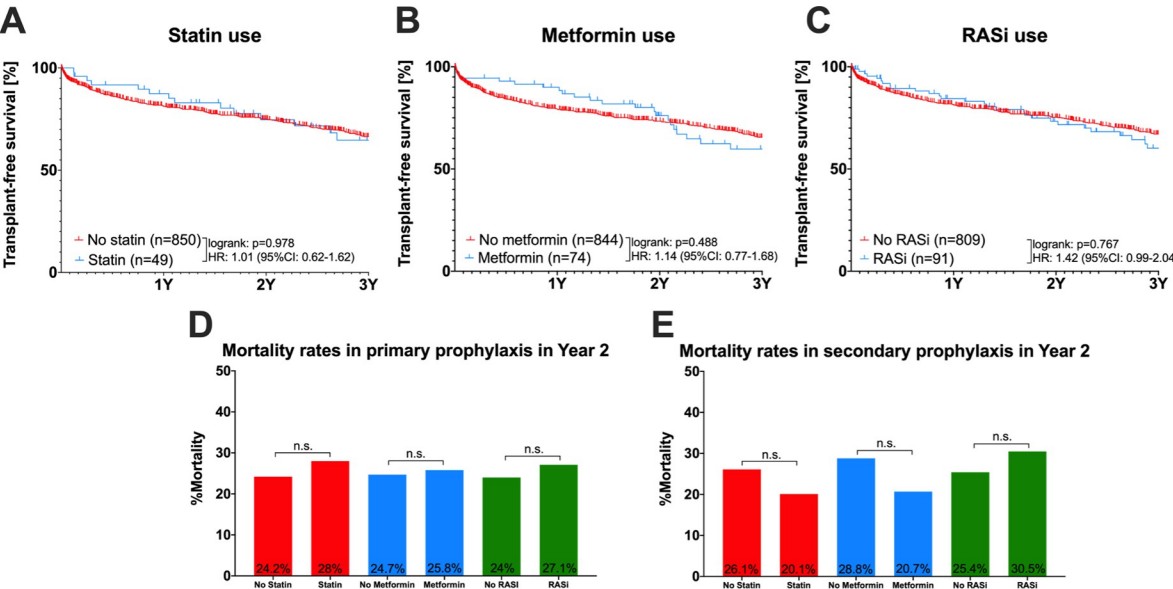

**Fig 3. Mortality rates.** Kaplan-Meier curves showing transplant-free survival for all patients with comedication (statin, metformin or RASi) or non-comedication (A-C). Bar chart of mortality rates in year 2 with comedication or non-comedication for primary prophylaxis and secondary prophylaxis (D,E). n (total numbers), 1Y (Year 1), 2Y (Year 2), 3Y (Year 3), HR (Hazard-Ratio); 95% CI (95% confidence interval); n.s. (not significant); statistical comparisons were performed by log-rank tests.

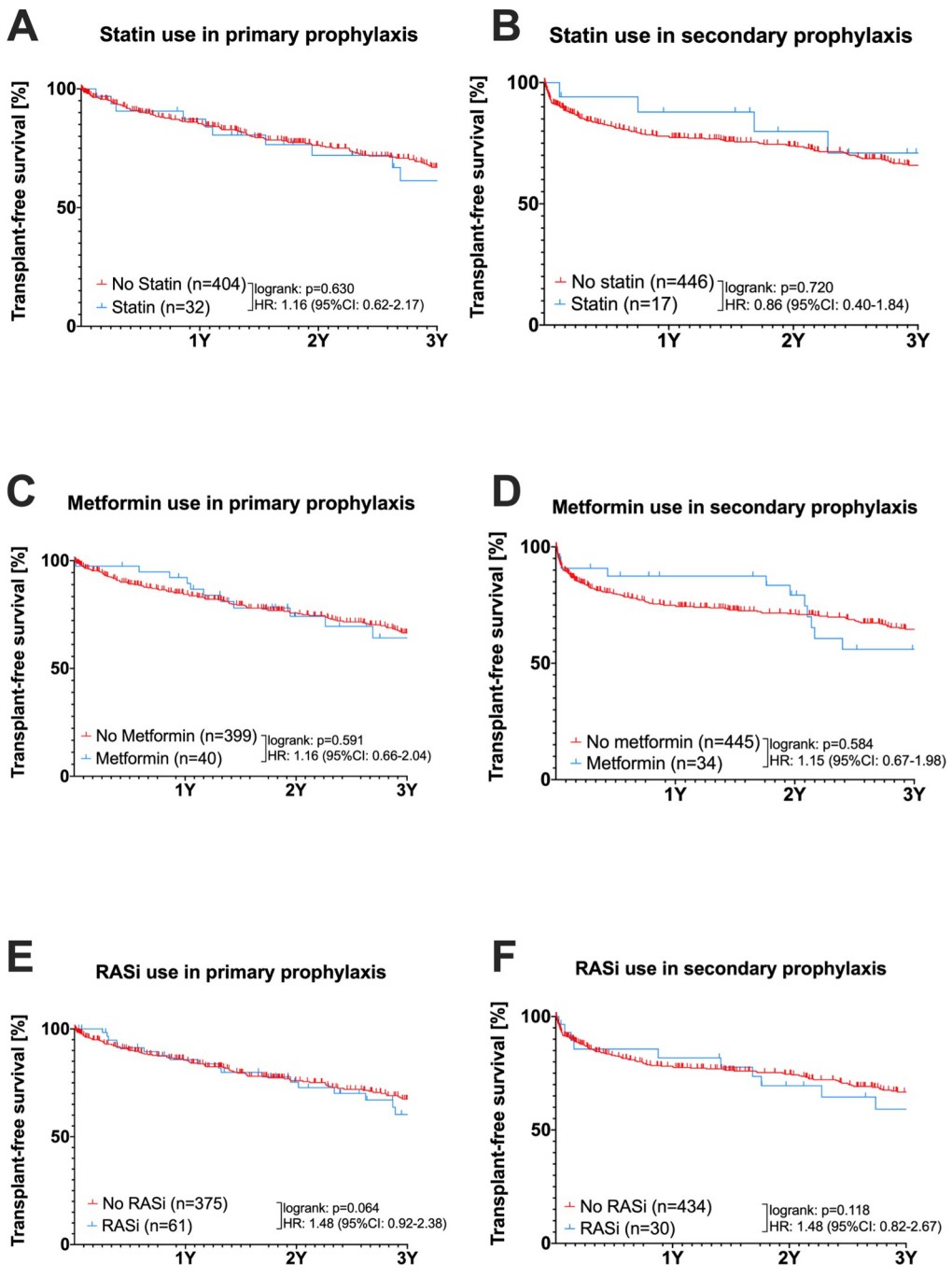

**Fig 4.** Kaplan-Meier mortality curves with comedication (statin, metformin or RASi) or non-comedication for primary and secondary prophylaxis (A-F). n (total numbers), 1Y (Year 1), 2Y (Year 2), 3Y (Year 3), HR (Hazard-Ratio); 95% CI (95% confidence interval); n.s. (not significant); statistical comparisons were performed by log-rank tests.

primary prophylaxis (HR: 1.16; 95%CI: 0.66–2.04; log-rank p = 0.591) and secondary prophylaxis (HR: 1.15; 95%CI: 0.67–1.98; log-rank p = 0.584). Furthermore, a similar rate of transplant-free survival was shown in patients using additional RASi in primary prophylaxis (HR:

1.48; 95%CI: 0.92–2.38; log-rank p = 0.064) and secondary prophylaxis (HR: 1.48; 95%CI: 0.82–2.76; log-rank p = 0.118).

### Bleeding for patients with combined treatment with NSBB and co-medication (statin, metformin or RASi) or non-co-medication in primary and secondary prophylaxis (Fig 5)

In *Fig 5*, patients were subdivided also according to primary and secondary prophylaxis and the impact of combined treatment with NSBB and co-medication on (re-)bleeding was analyzed. In the combination therapy of NSBB and statins, there was no difference in bleeding events observed between primary (HR: 3.37; 95%CI: 0.66–17.39; log-rank p = 0.146) and secondary prophylaxis (HR: 0.39; 95%CI: 0.13–1.16; log-rank p = 0.216). Additionally, the additional administration of NSBB, either with metformin (primary prophylaxis: HR: 0.81 95%CI: 0.28–2.38; log-rank p = 0.703; secondary prophylaxis: HR: 1.35 95%CI: 0.62–2.93; log-rank p = 0.203) or RASi (primary prophylaxis: HR: 1.09 95%CI: 0.41–2.88; log-rank p = 0.865; secondary prophylaxis: HR: 0.47 95%CI: 0.18–1.18; log-rank p = 0.638), did not show any advantage in terms of bleeding events in both primary and secondary prophylaxis.

## Discussion

In this real-life cohort study, we assessed (re)bleeding and mortality rates using additional co-medication with a statin, metformin, angiotensin-converting-enzyme inhibitors, or angiotensin II receptor blockers (RASi) in cirrhotic patients with esophageal varices treated at two major liver units. These medications were primarily utilized for the treatment of specific medical comorbidities such as dyslipidemia, diabetes, and arterial hypertension, respectively.

To our knowledge, this is the first retrospective study examining and summarizing the clinical impact of statin, metformin and RASi use, which may have a positive effect on portal hypertension in patients under primary and secondary bleeding prophylaxis for esophageal varices. However, in this multicenter real-life study including many patients, we could not reveal any benefits of additional therapy with metformin, statins, or ACE-I/ARB on the rate of variceal bleeding and mortality. The study could show that the intake of at least one comedication did not show a significant impact either on (re-)bleeding risk or mortality, even after adjustment for a higher Child-Pugh stage. It's also worth mentioning that as shown in Table 4, mortality rates were understandably significantly higher among patients classified as Child B/C and those who were older.

In the subgroup analyses, we subdivided the study cohort according to primary and secondary prophylaxis settings to accommodate the previously reported differences in clinical characteristics, current therapeutic approaches, and mortality among these two groups [5, 9, 29, 40]. However, our analyses did not provide any support for potential additional clinical effects of the evaluated drugs.

We observed a significantly lower rate of co-medication intake in patients under secondary prophylaxis compared to primary prophylaxis. A plausible reason for this could be that these patients are in a more severe state of illness, and have, for instance, significantly higher serum creatinine levels compared to those in the primary prophylaxis group (0.90mg/dL IQR 0.37 vs. 0.82mg/dL IQR 0.31, p = 0.001). This difference underlines the potential influence of clinical decision-making processes on treatment approaches, particularly in the context of patients after a severe event of variceal bleeding. This is in line with the results of a previous study, highlighting that the guideline-recommended use of statins is frequently withheld from patients with liver disease and is associated with decreased survival [41]

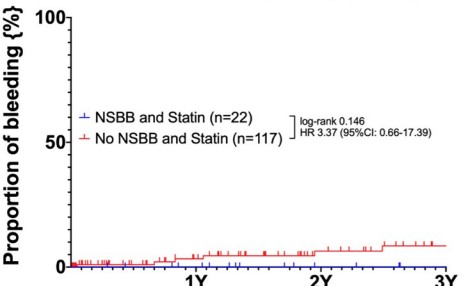

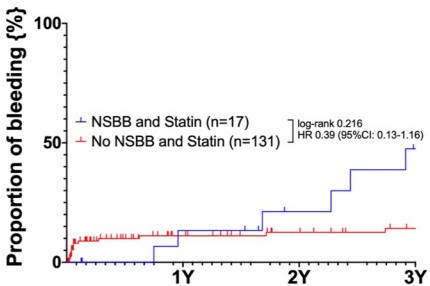

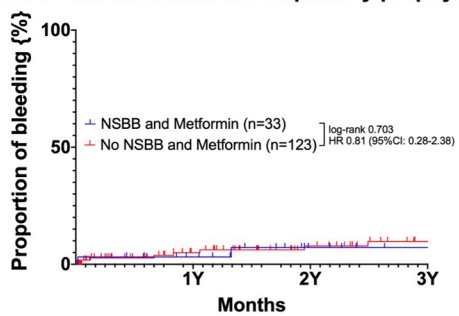

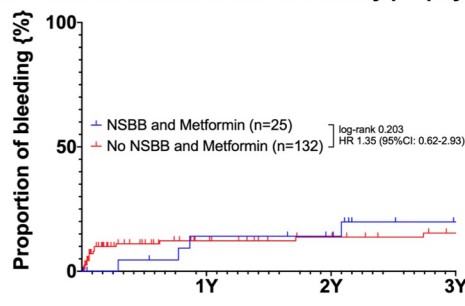

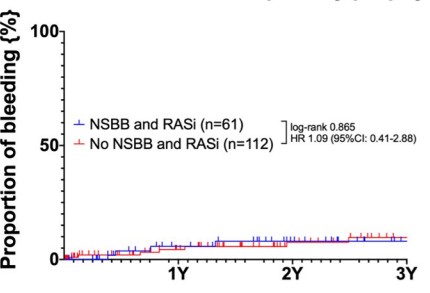

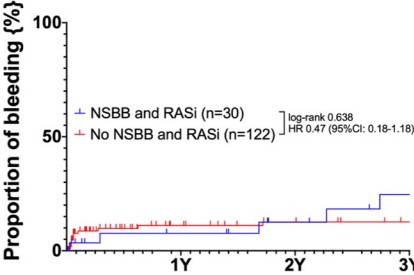

**Fig 5. Kaplan-Meier (re-)bleeding curves with NSBB and comedication (statin, metformin or RASi) or non-comedication for primary and secondary prophylaxis.**

Similarly, the rate of recommended NSBB use (26) was also low in our study, particularly in the context of secondary prophylaxis. This could be due to non-recorded contraindications or intolerance to NSBBs, but also to preferences of the respective treating physicians–as we have previously documented for Austria [42, 43]. However, the impact of the synergistic effect of NSBB to individual co-medications (statin, metformin, or RASi) on bleeding has no significant effect (see Fig 5).

Interestingly, in recent years, an increasing number of reports have shown the positive effects of statins in patients with liver cirrhosis, including reductions in the rates of hepatic decompensation, infections, and mortality [44] However, observational studies on statins have potential biases, which have been widely recognized and make it difficult to accurately determine the effects of statins on cirrhotic patients, as factors other than statins themselves may be affecting the results (e.g. loss of follow-up, incidence of cardiovascular death and/or cancer) [28].

In accordance with our results, the BLEPS trial, a randomized study in which 158 patients in secondary prophylaxis were additionally treated with simvastatin, did not show a significant reduction of rebleeding rates (HR: 0.86, p = 0. 583) [27] However, a significant decrease in overall mortality in patients with Child-Pugh A and B cirrhosis was shown (HR: 0.39, p = 0.030); yet this finding was not completely understood according to the authors [27] Another study from Mohanty et al. showed a greater effect of statins on quantitative reduction of mortality than on decompensation, suggesting that the benefits of statins regarding survival are not only related to cirrhosis [45]. In our study, the addition of statins in patients with portal hypertension did not result in a higher survival rate across the overall study cohort, nor in the primary (HR: 1.16, p = 0.630) and secondary prophylaxis (HR 0.86, p = 0.720) subgroups. However, these results could be biased by a higher rate of comorbidities in statin users, which may impact life expectancy. It is well known that the prevalence of comorbidities and the number of medications increase in elderly patients [46].

Hepatic venous pressure gradient (HVPG), representing the gold standard in the diagnostic assessment of portal hypertension, was not evaluated in this retrospective study, which represents a limitation. However, previous data showed that administration of statins leads to only moderate to non-reduction of HVPG, which is associated with the redistribution of blood flow from the collaterals to the liver as opposed to splanchnic vasodilatation [21, 47].

Overall, the most important limitation of this study is its retrospective design. Groups were not randomized and did not receive statin, metformin, or RASi therapy to treat the complications of liver cirrhosis, but rather to treat comorbidities, e.g. hypercholesterolemia, diabetes mellitus, heart failure, post-myocardial infarction, etc. Hence, it should be considered that the mortality of patients with additional medication intake is already increased due to metabolic and cardiovascular comorbidities, potentially attributing to the observed non-advantage of the assessed co-medications concerning survival. Another important limitation is the missing information on cumulative drug exposure prior to study inclusion. Furthermore, we only have exact compliance records of 37.3% of using NSBB. However, 87.2% have consistently taken NSBB over an extended period, which indicates a high level of compliance. Another important point is that the study primarily focused on collecting data on alcohol consumption at baseline as the identified cause of cirrhosis. Ongoing alcohol abuse throughout the follow-up period was not collected.

In summary, regarding the hard endpoints of this real-life multicenter study, no significant impact of statins, metformin, or RASi on bleeding events and survival was observed. Additional subgroup analyses in patients receiving primary or secondary prophylaxis revealed also no clear beneficial effects on variceal (re-)bleeding nor mortality. However, future studies using well-matched control groups regarding comorbidities–ideally of randomized design, should explore the effects of statins, metformin, and RASi on clinical endpoints in patients with cirrhosis and portal hypertension. Furthermore, an investigation of the impact of statins, metformin, and RASi specifically in patients with liver cirrhosis and "early" portal hypertension, e.g. before varices have bled would be an interesting focus for a separate study.

## Author Contributions

**Conceptualization:** Nikolaus Pfisterer, Michael Schwarz, Caroline Schwarz, Florian Putre, Lukas Ritt.

**Data curation:** Nikolaus Pfisterer, Michael Schwarz, Caroline Schwarz, Florian Putre, Lukas Ritt, Florian Riedl, Lukas Hartl, Mathias Jachs, Mattias Mandorfer, Thomas Reiberger.

**Formal analysis:** Nikolaus Pfisterer, Mattias Mandorfer, Thomas Reiberger.

**Supervision:** Christian Madl, Michael Trauner, Thomas Reiberger.

**Validation:** Thomas Reiberger.

**Writing – original draft:** Nikolaus Pfisterer.

**Writing – review & editing:** Michael Schwarz, Mathias Jachs, Christian Madl, Michael Trauner, Thomas Reiberger.

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
