## [Decision Letter · Decision Letter 0]

9 Feb 2024

PONE-D-23-35530Statins, metformin, and RAS inhibitors did not reduce variceal bleeding risk and mortality in a large, real-life cohort of patients with cirrhosisPLOS ONE

Dear Dr. Reiberger,

Thank you for submitting your manuscript to PLOS ONE. After careful consideration, we feel that it has merit but does not fully meet PLOS ONE’s publication criteria as it currently stands. Therefore, we invite you to submit a revised version of the manuscript that addresses the points raised during the review process.

We look forward to receiving your revised manuscript.

Kind regards,

Pavel Strnad

Academic Editor

PLOS ONE

2. Please amend either the title on the online submission form (via Edit Submission) or the title in the manuscript so that they are identical.

4. Please include the reference section of your manuscript. 

Reviewers' comments:

Reviewer's Responses to Questions

**Comments to the Author**

1. Is the manuscript technically sound, and do the data support the conclusions?

Reviewer #1: Yes

Reviewer #2: Yes

2. Has the statistical analysis been performed appropriately and rigorously? 

Reviewer #1: N/A

Reviewer #2: Yes

3. Have the authors made all data underlying the findings in their manuscript fully available?

Reviewer #1: Yes

Reviewer #2: Yes

4. Is the manuscript presented in an intelligible fashion and written in standard English?

Reviewer #1: Yes

Reviewer #2: Yes

5. Review Comments to the Author

Reviewer #1: Pfisterer et al conducted a retrospective study investigating the use of statins, metformin, ACE-inhibitors, and angiotensin-II receptor blockers, which have been suggested to have a beneficial effect in the context of portal hypertension. They examined the impact of their intake at the time of primary or secondary prophylaxis of esophageal varices on cirrhosis endpoints, specifically variceal (re-)bleeding and mortality. In the following, I will refer to the patients taking these medication as "comed"-group.

The strength of the study lies in its real-life setting, conducted in two major liver units with a large cohort and statistically robust methods. The paper is well-written, and the authors employed well-considered inclusion and exclusion criteria. However, there are major points that need addressing:

Timepoint of Comparison: In my opinion, the authors should have selected a different timepoint for comparison. At the time of inclusion, patients already exhibited clinically significant portal hypertension, indicated by the presence of high-grade varices or a bleeding event in the secondary prophylaxis group. The consequences of portal hypertension likely overrode the effects of comeds at this stage. Moreover, it would be beneficial to know, how long before baseline these medications were taken. The time from the start of intake to baseline should be included in the statistics or, if unavailable, this should be mentioned in the limitations. A more appropriate timepoint for comparison might be 1 or 2 years after beginning therapy with these comeds. Do the authors possess this data? Perhaps, using this timepoint of analysis, it might not even be necessary to compare between primary and secondary prophylaxis groups.

Comparison Between Groups: In the primary prophylaxis group, 13.2% of patients experienced a first bleeding event, and 36.8% died. In the secondary prophylaxis group, 30% had further bleeding events, and 49.1% died. Is the comparison significant between the two groups? Did the authors compare the number of patients with comeds vs. no-comeds in the respective groups? The authors present results only within the two groups.

Confounders: Some patients in the comed and no-comed groups had ascites, a known major factor related to prognosis and potentially introducing bias. Did the authors perform sub-analyses for patients with and without ascites at baseline, or stratified by CTP-class (e.g., CTP A vs. B/C)? If the number of patients in the respective comed groups is too small, perhaps combining all patients in one group for analysis could be considered.

Comorbidities: One of the major weaknesses of the study, as acknowledged by the authors, is that comeds were not intentionally used for liver purposes, suggesting that patients with comeds were sicker at baseline. This could be a reason why no beneficial effect was observed for patients taking these comeds. At least there should be some statistical adjusting between the comed and not no-comed groups before comparing, or as mentioned before, substratifications of CTP class should be shown and then compare decompensation events such as bleeding or mortality.

Role of NSBB: The role of NSBB in this study is not clear to me. The authors state the numbers of patients receiving NSBB at baseline, but details about continuation/discontinuation after primary or secondary prophylaxis and the duration of use before are missing. It is now widely known that the early use of NSBBs prevents decompensation events. Additionally, it might be worthwhile to explore the synergistic effect of beta-blockers and comeds on the stated endpoints.

Minor Points:

- The authors mention that patients in whom co-medication could not be reliably assessed were not considered for this analysis. How many patients fall into this category? It should be stated in the manuscript.

- The manuscript lacks p-values in the statistics. If there is no statistical difference between the all groups, it should be explicitly stated in the manuscript.

In summary, while I commend the authors for presenting these real-life data, a more thorough confounder analysis is necessary before making this strong statement. Consideration of a different timepoint for comparison might enhance the robustness of the study.

Reviewer #2: In their comprehensive study, Pfisterer et al. analysed the possible beneficial effect of statins, metformin, angiotensin-converting-enzyme inhibitors and angiotensin II receptor blockers (RASi) in primary and secondary prophylaxis of variceal bleeding.

The authors observed that in clinical practice, variceal bleeding and mortality rates of cirrhotic patients were not reduced by co-medication with statins, metformin or RASi. Nevertheless, it is recommended to use these co-medications by indication, as they may still have beneficial effects on non-bleeding complications in patients with liver cirrhosis.

*As the authors stated, the major limitation of this study is its retrospective character. Because of that, the groups were not randomized and did not receive statin, metformin, or RASi therapy to treat the complications of liver cirrhosis but rather to treat the patient´s comorbidities.

* In the manuscript, it is not stated if the authors had analysed possible ongoing alcohol abuse, especially in patients with alcoholic liver disease, which increases the risk of variceal bleeding. It would be interesting if the authors could divide patients into cohorts of ongoing alcohol abusers and non-abusers. If the data are not available, a comment on ongoing alcohol use and the impact of cessation of alcohol intake would be valuable.

6. PLOS authors have the option to publish the peer review history of their article (what does this mean?). If published, this will include your full peer review and any attached files.

Reviewer #1: No

Reviewer #2: **Yes: **Mikolas Holinka

---

## [Author Response · Author response to Decision Letter 0]

26 Mar 2024

We would like to express our gratitude for the acceptance and thorough review of our study on bleeding and mortality rates in a large, real-world cohort of patients with cirrhosis that focused on the impact of co-medication with statins, metformin, and RAS inhibitors. We have taken this opportunity to revise our study and considered all valuable comments and suggestions received from the Editors and the Reviewers. We have collected additional data and performed detailed analyses that ultimately, further strengthen our conclusions. Specifically, we have:

• assessed follow-up duration, bleeding events and mortality separately for the primary prophylaxis group and the secondary prophylaxis group, while still comparing the intake of comedication within these subroups.

• utilized a Cox regression model to adjust for Child-Pugh stage and age, showing that the intake of at least one comedication did not significantly affect (re-)bleeding risk or survival outcomes.

• performed additional analyses on the effects of individual comedications on bleeding when added on top of NSBB treatment and found that, in both primary and secondary prophylaxis, the synergistic effect of NSBB and all comeds on decompensation, such as (re-)bleeding, has no significant effect.

• Revised our discussion section by considering the important suggestions pointed out by the reviewers, such as the lack of systematic assessment of ongoing alcohol consumption.

Please find our point-by-point response below. Thank you for considering our revised article for publication in PLOS ONE. With kind regards on behalf of all co-authors, Dr. Nikolaus Pfisterer and Prof. Thomas Reiberger

---

## [Decision Letter · Decision Letter 1]

15 Apr 2024

Statins, metformin, and RAS inhibitors did not reduce variceal bleeding risk and mortality in a large, real-life cohort of patients with cirrhosis

PONE-D-23-35530R1

Dear Dr. Reiberger,

We’re pleased to inform you that your manuscript has been judged scientifically suitable for publication and will be formally accepted for publication once it meets all outstanding technical requirements.

Kind regards,

Pavel Strnad

Academic Editor

PLOS ONE

Additional Editor Comments (optional):

Reviewers' comments:

Reviewer's Responses to Questions

**Comments to the Author**

1. If the authors have adequately addressed your comments raised in a previous round of review and you feel that this manuscript is now acceptable for publication, you may indicate that here to bypass the “Comments to the Author” section, enter your conflict of interest statement in the “Confidential to Editor” section, and submit your "Accept" recommendation.

Reviewer #1: (No Response)

Reviewer #2: All comments have been addressed

2. Is the manuscript technically sound, and do the data support the conclusions?

Reviewer #1: Yes

Reviewer #2: Yes

3. Has the statistical analysis been performed appropriately and rigorously? 

Reviewer #1: Yes

Reviewer #2: Yes

4. Have the authors made all data underlying the findings in their manuscript fully available?

Reviewer #1: Yes

Reviewer #2: Yes

5. Is the manuscript presented in an intelligible fashion and written in standard English?

Reviewer #1: Yes

Reviewer #2: Yes

6. Review Comments to the Author

Reviewer #1: The authors have adequately addressed all concerns. I commend them for acquiring new data and perform additional statistical analyses and adjusting for confounders that make the main message of the paper more robust. Regarding the topic itself, I still believe that comparing patients at the time of re-/bleeding, probably with the presence of cardiovascular comorbidities, may not be the optimal timepoint for comparison, resulting in somewhat predictable results. However, as the authors point out, there is a scarcity of data on this topic, and their study will contribute important information. They also acknowledge the limitations, particularly the retrospective nature of the study and the absence of cumulative dosage information for statin/metformin/RASi up to the baseline timepoint.

Reviewer #2: Dear authors thank you for addressing all of my comments. I am recommending this manuscript for acceptance.

7. PLOS authors have the option to publish the peer review history of their article (what does this mean?). If published, this will include your full peer review and any attached files.

Reviewer #1: No

Reviewer #2: **Yes: **Mikoláš Holinka

---

## [Editor Report · Acceptance letter]

7 May 2024

PONE-D-23-35530R1 

PLOS ONE

Dear Dr. Reiberger, 

I'm pleased to inform you that your manuscript has been deemed suitable for publication in PLOS ONE. Congratulations! Your manuscript is now being handed over to our production team.

Kind regards, 

on behalf of

Dr. Pavel Strnad 

Academic Editor

PLOS ONE